# Milan's forgotten epidemic of summer 1629, a few months before the last great plague: An investigation into the possible cause

**Massimo Galli**[1] *, **Letizia Oreni**[1], **Anna Lisa Ridolfo**[1], **Angelo Formenti**[1], **Ester Luconi**[2], **Patrizia Boracchi**[2], **Spinello Antinori**[1], **Elia Biganzoli**[2], **Folco Vaglienti**[3]

1 Unit of Infectious Diseases, Department of Biomedical and Clinical Sciences, Università degli Studi di Milano, Milan, Italy, 2 Unit of Medical Statistics, Biometry and Epidemiology, Department of Biomedical and Clinical Sciences, and DSRC, Università degli Studi di Milano, Milan, Italy, 3 Department of History and Historical Documentation Sciences, Università degli Studi di Milano, Milan, Italy

* massimo.galli@unimi.it

## Abstract

An epidemic not attributable to plague caused thousands of deaths in Milan in the summer of 1629, a time of war and famine that immediately preceded the even more fatal Great Plague of 1630 that killed an estimated ten of thousands of people. The 5,993 deaths of 1629 recorded in the *Liber Mortuorum* of Milan (a city with an estimated population of 130,000 inhabitants at the time) were 45.7% more than the average number recorded between 1601 and 1628. Registered deaths peaked in July, and 3,363 of the deaths (56,1%) were attributed to a febrile illness which, in most cases (2,964, 88%), was not associated with a rash or organ involvement. These deaths involved 1,627 males and 1,334 females and occurred at a median age of 40 years (range 0–95). In this paper, we discuss the possible cause of the epidemic, which may have been an outbreak of typhoid fever.

## Introduction

In the year 1629, the war of succession in Mantua was promising further troop movements across the Alps and, with them, a new plague epidemic in Northern Italy [1]. The discontent caused by the difficult economic conditions of the population, which was oppressed by the Spanish domination and the taxes imposed to support the war, led to a popular uprising in November 1628 [2, page 25] that ended in December with the hanging of the four people identified as its leaders. From the beginning of 1629, thousands of impoverished peasants fled to Milan in search of a livelihood [2–7] and, in the spring of the same year, people began to drop dead in the street, "some with the roots of herbs, others with bran in their mouths" [3]. From the end of May, all beggars were confined to the Lazaretto of St. Gregory, although this was against the advice of the *Magistrato di Sanità* (the government authority responsible for public health) [8], which feared it could generate a contagious focus [3]. Mortality in the Lazaretto rapidly increased, and soon reached an average of 70–80 deaths per day [9]. In June, all the less debilitated inmates were discharged and, according to the contemporary physician and chronicler Alessandro Tadino [3], the disease soon spread throughout the city ("*si comunicò il male*").

**Data Availability Statement:** All relevant data are within the paper and its Supporting Information files.

**Funding:** The authors received no specific funding for this work.

**Competing interests:** The authors have declared that no competing interests exist.

In a handwritten note of 1632, another contemporary physician, Senatore Settala, called the disease "... *quel morbino di San Gregorio* ... *che è non solo febre maligna ma pestilentiale et attacaticia et quasi peste* (i.e., "that little *morbus* of St. Gregory, which is not only a malignant fever but pestilential, contagious, and almost like a plague") [10]. According to Tadino, there were 8,750 deaths in Milan between January and August 1629 [3], a figure that was substantially confirmed by the Resident of the Grand Duchy of Tuscany, who wrote in July that "seven thousand souls have died here since January, mostly people of low rank" [7]. Moreover, the Resident of the Republic of Venice sent a dispatch to the *Serenissima* on 1 August in which he claimed that the number of deaths was as high as 14,000 in just two months [11].

The terrible Great Plague that occurred in Milan just a few months later in 1630 has almost erased the memory of the previous summer's epidemic, the cause of which remains unknown. The aim of this study was to analyse the deaths listed in 1629 in the Milan *Mortuorum libri* (MML), a series of registers indicating the causes of deaths in the city (excluding those occurring in convents, hospitals, and the Lazzaretto) that was started in 1450 and is one of the oldest and most complete systematic collections of public health data in pre-industrial Europe [12].

## Methods

The MML of 1629 was consulted at the *Archivio di Stato* in Milan [13] and used to construct a database containing the demographic data of the deceased (name, surname, age, gender, titles or social status) and the main cause of death and other contributing illnesses as described by the city's appointed medical officers (S1 File). By order of the health authority, MML registration was extended to all deaths occurring in the city (in private homes and inns, or on the street), but not those occurring in hospitals, convents, or the Lazzaretto; it also became mandatory to obtain a burial permit [12].

The monthly number of deaths in 1629 was compared with the number occurring in the period 1601–1628, as reported in Giuseppe Ferrario's 1840 list of all the deaths recorded in the MML each year by calendar month [14]. The distribution of the deceased by age and gender was compared using the Kolmogorov-Smirnov test.

Sennert's 1608 treatise [15] was used as a reference to interpret the meaning attributed to the different types of fever associated with the deaths by the drafters of the death certificates. Accordingly, deaths due to fever or in which fever was involved were categorised as follows: a) fever without a description of organ involvement; b) fever associated with gastrointestinal symptoms; c) fever associated with respiratory symptoms; d) fever with rash; and e) *febre etica*, a term describing the chronic/long-lasting fever often associated with phthisis or long-lasting/debilitating diseases [15]. The deaths attributed to *epilepsia*, a term that may have been used to describe infantile febrile convulsions or infantile weakness without any associated fever, epilepsy, or epileptic disease in people of any age were not included in the analysis of deaths caused by febrile illness.

The weekly number of deaths due to fever was recorded, as were the personal data of the people whose deaths were attributed to the different types of fever (age, sex, and the number of days between symptom onset and death when available), and the name and professional position (non-graduate surgeon or graduate doctor) of the health officer certifying each death.

Finally, the chi-squared test was used to compare the percentage of deaths due to fever without organ involvement in males and females stratified by age.

## Results

The 1629 MML lists 5,997 cases of death but, as four proved to be duplicates or erroneous transcriptions, the actual number was 5,993, of which 211 were recorded in the city's three prisons,

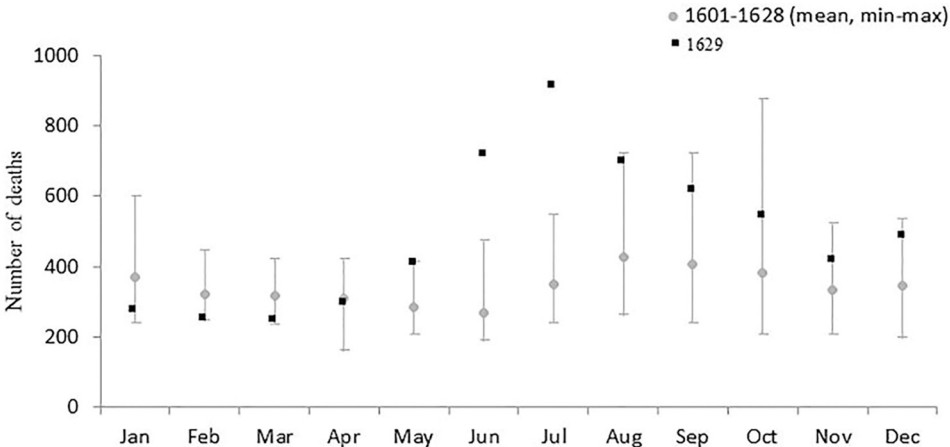

**Fig 1. The number of deaths recorded in Milan's *Liber Mortuorum* for each month of 1629, and the mean (min-max) monthly number of deaths in the period 1601–1628.**

171 involved beggars found dead in the street, and 37 "foreigners" (including nine coming from other Italian cities and 11 Spaniards). Most of the entries (71%) were made by two *cerusici* (non-graduate surgeons) specifically appointed to inspect corpses; the others were written by 33 *physici collegiati* (graduate doctors). The number of deaths recorded in 1629 was 45.7% higher than the annual average of 4,113 recorded during the period 1601–1628. A significant excess of deaths was recorded in June and July: respectively 718 and 911 deaths *vs* the mean numbers of 268 (range:189–473) and 349 (range: 240–547) deaths recorded in the same months during the period 1061–1628 (Fig 1).

The deceased included 3,072 males and 2,788 females; gender could not be deduced in 133 cases (2.2%), mainly involving people found dead in the street whose names were unknown. Their median age at the time of death was 40 years (range 0–107); this information was not available in 200 cases (3.3%). As shown in Fig 2, there were no significant differences in the gender distribution of the deaths among age groups.

A total of 3,363 deaths were attributed to or associated with fever. Table 1, panel A, shows the number of deaths attributed to the different patterns of fever. Fever without a description of organ involvement was the reported cause of death in 2,964 cases, and fever associated with gastrointestinal (*cum fluxo*) or respiratory symptoms (*cum catarro*) was reported in respectively 141 and 144 cases. Excluding six deaths due to *variola* (smallpox) between the ages of two months and 21 years, there were only 12 cases of death due to fever with rash (age range 14–65 years). The 102 deaths associated with *febre etica* occurred in much younger subjects than those with the other types of fever, regardless of organ involvement, and were associated with respiratory symptoms in five cases, two of which were also associated with hemoptysis (*cum sputo sanguinis*). *Febre etica* was also reported in the case of two deaths mainly attributed to *phtisi*.

The following definitions of fever without any specific reference to organ involvement were found in the MML: *febre acuta* (1,793 cases), *febre* (320), *febre longa* (315), *febre continua* (218), *febre acuta maligna* (203), *febre maligna* (74), *febre terzana* (15), *febre peracuta* (14), and *febre frenetica* (12) (Table 1, panel B). In some cases, including 7.9% of the deaths due to *febre acuta* and 38.4% of those due to *febre acuta maligna*, the time interval between symptom onset and death was recorded: in both cases, the median interval was ≤10 days. The *cerusici* and *phisici collegiati* tended to use different terms to describe this type of fever. The deaths due to

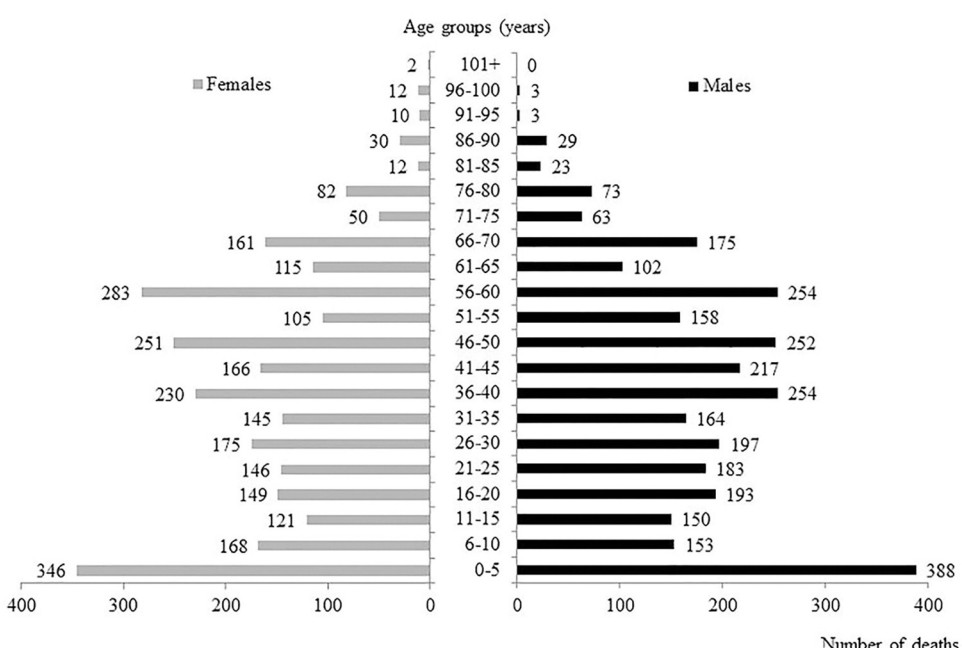

**Fig 2. Total number of deaths recorded in 1629, stratified by gender and age group at the time of death (age not available in 200 cases).** The number of deaths in each group is shown next to the horizontal bar.

*febre longa* (long fever) or *febre* without any other specification were prevalently recorded by *cerusici*, and all the diagnoses of *febre frenetica* were formulated by only one of the two *cerusici*. All of the diagnoses of *febre peracuta* were made by *phisici collegiati* (all but one by the same doctor) and, although they made only about 40% of the diagnoses of death due to fever without organ involvement, *phisici collegiati* made 60% of the diagnoses of *febre continua*.

Fig 3 shows the weekly distribution of deaths associated with the different patterns of fever. The number of deaths attributed to fever without any reported organ involvement began to increase in week 19 of 1629 and peaked in week 29; the number of deaths due to fever with gastrointestinal symptoms showed a similar temporal trend, whereas those due to fever with respiratory symptoms were more evenly distributed throughout the year.

One hundred and forty-four (84.2%) of the 171 deaths of beggars found dead in the street were recorded between week 24 and week 28 (from mid-June to mid- July), and the entries covering the deaths of 128 included up to seven people at a time, without any indication of their age, sex, or cause of death.

The distribution of deaths due to different types of fever without any description of organ involvement was substantially comparable, with most of the deaths being recorded during the summer months (S1 Fig); however, the distribution of the deaths associated with *febre etica* was not seasonal.

Overall, deaths due to fever without any description of organ involvement were relatively more frequent among males (53% *vs* 47.8%; p<0.001), particularly in the age stratum of 41–45 years (67.3% *vs* 54.8%; p = 0.01) (Fig 4).

## Discussion

The available data show that Milan was hit by a major epidemic in the summer of 1629 that caused a 60% increase in the number of recorded deaths in comparison with 1628. The officially recorded deaths of 171 beggars (2.9% of the total for the year) are not enough to explain

**Table 1. Characteristics of the deaths attributed to different patterns of fever (panel A), and different types of fever without any specific reference to organ involvement (panel B).**

| Panel A | | | | | | |
|---|---|---|---|---|---|---|
| Described patterns of fever | No. | Sex, n (%) | | Age, years | Days from symptom onset to death | |
| | | Males | Females | Median (range) | No. of cases | Median (range) |
| Fever without any described organ involvement | 2964[‡] | 1627 (54.9) | 1334 (45.1) | 40 (0–95) | 260 | 10 (1–181) |
| Fever associated with gastrointestinal symptoms | 141 | 78 (55.3) | 63 (44.7) | 50 (2–90) | 13 | 14 (3–181) |
| Fever associated with respiratory symptoms | 144 | 70 (48.6) | 74 (51.4) | 60 (1–95) | 10 | 7 (3–30) |
| *Febre etica*[◇] | 102 | 44 (43.1) | 58 (56.9) | 20 (2–74) | 6 | 151 (91–181) |
| Fever with rash | 12 | 7 (58.3) | 5 (41.7) | 41 (14–65) | - | - |
| Panel B | | | | | | | |

| Different types of fever without any described organ involvement | No. (%) | Sex, No. (%) | | Age, years | Certifiers of deaths, No. (%) | | Days from symptom onset to death | |
|---|---|---|---|---|---|---|---|---|
| | | Males | Females | | *Cerusici* | *Physici collegiati* | | |
| | (n = 2964) | (n = 1627) | (n = 1334) | Median (range) | (n = 1766) | (n = 1197) | No. of cases | Median (range) |
| *Febre acuta* | 1793 (60.5)[‡] | 976 (60.0) | 814 (61.0) | 40 (0–95) | 1091 (61.8) | 702 (58.6) | 142 | 10 (2–30) |
| *Febre* | 320 (10.8) | 176 (10.8) | 144 (10.8) | 30 (0–90) | 222 (12.6) ** | 98 (8.2) | 2 | 11 (8–14) |
| *Febre longa* | 315 (10.6) | 151 (9.3) | 164 (12.3) * | 35 (1–90) | 287 (16.3) *** | 28 (2.3) | 2 | 121 (60–181) |
| *Febre continua* | 218 (7.4)[§] | 119 (7.3) | 99 (7.4) | 48 (0–90) | 86 (4.9) *** | 131 (10.9) | 27 | 10 (3–24) |
| *Febre acuta maligna* | 203 (6.8) | 133 (8.2) | 70 (5.2) * | 45 (4–93) | 7 (0.4) *** | 196 (16.4) | 78 | 9 (3–20) |
| *Febre maligna* | 74 (2.5) | 47 (2.8) | 27 (2.0) | 40 (5–70) | 55 (3.1) * | 19 (1.6) | 7 | 7 (4–15) |
| *Febre terzana* | 15 (0.5) | 8 (0.5) | 7 (0.5) | 60 (21–75) | 6 (0.3) | 9 (0.8) | 2 | 37 (14–60) |
| *Febre peracuta* | 14 (0.5) | 9 (0.6) | 5 (0.4) | 49 (25–70) | 0 (0) *** | 14 (1.2) | - | - |
| *Febre frenetica* | 12 (0.4) | 8 (0.5) | 4 (0.3) | 30 (20–70) | 12 (0.7) * | 0 (0) | - | - |

‡Gender missing in three cases.

◇Definition referring to fever associated with phthisis or chronic debilitating diseases (ref. 14)

‡Gender missing in three cases.

§Certifier missing in one case.

*p-value <0.05

**p-value <0.001

***p-value <0.0001

the extent of this increase and, given that the number of deaths estimated by the chroniclers of the time is much larger than that recorded in the MML, it is unlikely that the number of unrecorded deaths in lazarettos and hospitals would have been enough to make up the difference. It is therefore likely that the deaths caused by the *morbino* were perceived as being more numerous than they actually were. The fact that, in addition to the deaths of non-residents occurring on the city's streets, the register also recorded the deaths of 37 other "foreigners" (including 11 Spaniards who were probably garrison soldiers and their families) tends to exclude the possibility that there are omissions other than the deaths of people dying in convents, hospitals and in the Lazzaretto, as required by the regulations governing the MML.

Although contemporary chroniclers emphasised the role of migrants in spreading the epidemic, the increase in the number of incident cases of death due to fever started in week 19,

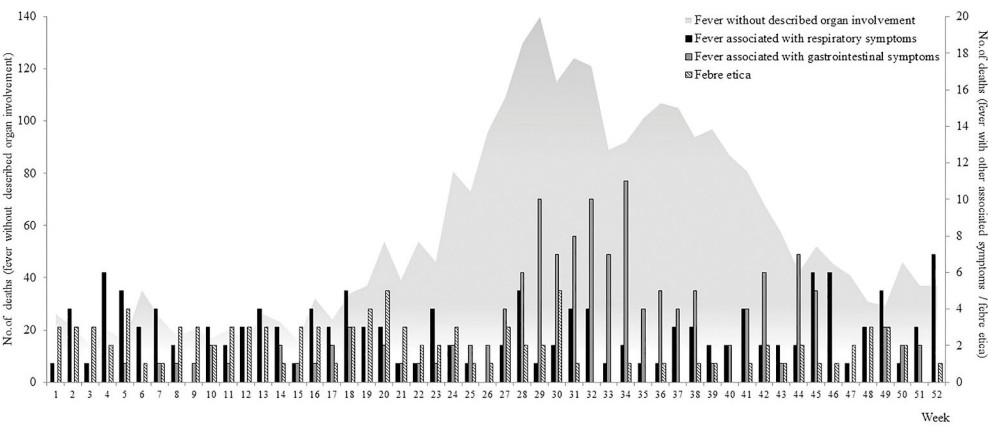

**Fig 3. Weekly distribution of the deaths attributed to fever without any described organ involvement in comparison with the distribution of deaths due to fever with gastrointestinal symptoms, fever with respiratory symptoms, and** *febre etica.*

and more than 80% of the deaths of beggars found dead in the street were recorded between week 24 and week 28 (presumably after their release from the Lazaretto of St. Gregory). It is also interesting to note that the MML describes such beggars as 'those of St. Gregory' (*ex illis Sancti Gregorii*) only from the second half of June.

The terms used to define the different types of fever followed the classification based on the theory of humours in only a minority of cases, and only two of the four types of putrid fever contemplated by Galen [16] (*febre continua* attributed to a blood alteration, and *febre terzana* attributed to a bile alteration) are recorded as being associated with the deaths that occurred in 1629. The use of these terms implies an effort to describe the symptom(s) and pathogenesis of the disease causing the deaths, and so it is not surprising that *febre continua* was more

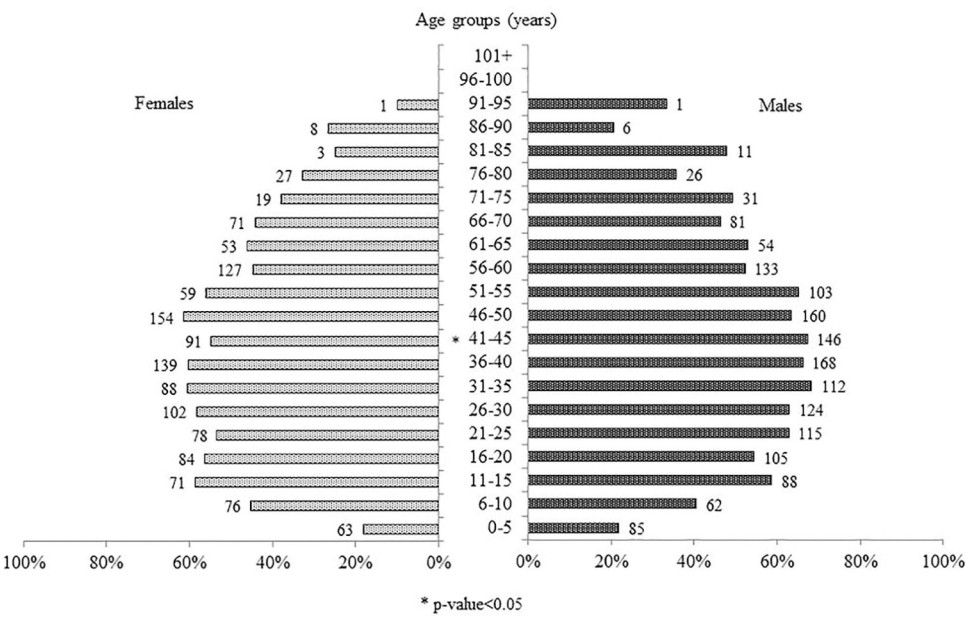

**Fig 4. Percentage and absolute number of deaths due to fever without any described organ involvement, stratified by gender and age.** The absolute number of deaths in each group is shown next to the horizontal bar.

frequently used by the *physici collegiati* than the two *cerusici*. However, neither of these types of fever differ from the other types of acute fever in terms of their distribution between the sexes or seasonality, the median age of the subjects at the time of death, or (when this information was available) the duration of the symptom(s) before death.

On the contrary, *febre etica* did not seem to be seasonal and, in comparison with the other types of fever, appeared at a much younger age and a longer time before death. The cases associated with phthisis or respiratory symptoms *cum sputo sanguinis* suggest a diagnosis of pulmonary tuberculosis. In his 1619 treatise on fevers, Sennert [15] described the long-lasting nature of this fever ("*in quibus intemperies febrilis quasi in habitum transivit*": Book III, Chap. I, page 333) and its association with phthisis ("*sæpissime febris talis excitatur ex pulmonum ulcere in phthisicis, qua vulgo plerique hecticam vocant*": Book II, Chap. XVI, page 208), and it is likely that the use of the term *febre etica* by Milanese public health officers in 1629 was in line with this medical tradition.

*Febre longa* was probably a diagnosis of convenience used (mainly by the two *cerusici*) to reinforce the exclusion of plague, which was believed to be fatal in a matter of days (Sennert, Book IV, Chap. IV, page. 421). Nevertheless, the deaths due to *febre longa* occurred at a similar median age and had a seasonal peak that like that of the large group of acute fevers without any description of organ involvement that caused most of the deaths in 1629. We have therefore included these cases in our analysis of deaths due to fever without organ involvement as a possible part of the epidemic.

Seventeenth century medical science certainly knew of the existence of malignant or pestilential fevers other than the plague (Sennert, Book IV, Chap. VII, pages 478–484), although there was much discussion about their nature. It is clear that the Milanese public health officers chose the terms used in their contemporary medicine to indicate the presence of an epidemic fever that was less severe and less lethal than the plague. The most correct and complete formulation would have been *febre acuta maligna*, which was used almost exclusively by the *physici collegiati*.

It is important to note that we did not find any reference any of the typical manifestations of plague such as buboes in the MML of 1629, but these can be found upon an even cursory reading of the 1630'registers. Historical documents [2, 3] precisely describe the dynamics of the plague epidemic before it entered Milan, including its crossing of the Alps with the imperial army in the autumn of 1629 and subsequent approach to the city, where unlikely it arrived before October, when the first case of a man dying of plague was reported [2]. As this death occurred in hospital, it was not recorded in the MML. However, it cannot be excluded that some of the cases of death attributed to malignant fever or acute malignant fever recorded in last months of 1929 were due to plague. On the other hand, the overall average duration of illness before death in these cases (7–9 days) was greater than the three or four days considered typical of plague [17, page 327]. Moreover, the *Protofisico Generale* (Chief Physician) of the Duchy of Milan, Lodovico Settala, published a treatise on plague in 1622 based on his experience of the 1576 epidemic [17], and many of the city's public health personnel were experienced in its empirical diagnosis [18]. Taken together, this seems to be sufficient to affirm that Milan's public health authorities were more than capable of recognising a case of plague, and that the *morbino* of Senatore Settala was something else. Finally, it is unlikely that the presence of a large plague epidemic could have been hidden for long, particularly from the watchful eyes of the Residents of the other Italian States.

Except for the six deaths attributable to smallpox, there are very few recordings of rashes or other cutaneous manifestations. This rarity of exanthematic manifestations strongly suggests that the *morbino* was not an outbreak of epidemic typhus, several episodes of which had occurred in Milan from the fifteenth century [19]. Epidemic typhus was described and

distinguished from plague by Fracastoro [20], who confirmed its exanthematous nature in 1547 [5]; furthermore, as a large outbreak had occurred in Florence in the winter of 1621 [21], it can be assumed that Milan's public health officers would have easily recognised it. Finally, epidemic typhus is more frequent during the winter, when heavy clothing favours the proliferation of lice and the transmission of *Rickettsia prowazekii* [22, 23], whereas the *morbino* was a summer epidemic.

The number of deaths reported in Milan between 1601 and 1629 was always higher in the summer [14] and, in 1629, their number increased at the end of May and reached a peak in July. In many ways, it seems that the *morbino* may have been an enteric fever, most probably typhoid fever which, although it is typically food- or water-borne, can also be transmitted by inter-human contacts [24]. The recurrent excess of registered deaths in Milan during the summer of every year from 1601 [14] suggests that enteric fevers were endemic, and that more or less extensive outbreaks occurred depending on the weather or other favouring events. Unfortunately, no data are available concerning these outbreaks other than their seasonality, and so they cannot be compared with the metrics of the 1629 epidemic.

As is now well known, the population maintenance of *Salmonella enterica* serotype Typhi (which was responsible for 76.3% of the enteric fevers reported worldwide in 2017) [25] is guaranteed by chronic carriers, who represent approximately 2–5% of the typhoid patients who fail to clear the infection fully within one year of recovery [26]. It is worth remembering that the first anatomical description of the ileus lesions later identified as being attributable to typhoid fever was made by Spigelius in 1624, when he was Professor of Anatomy at the University of Padua [27]. Males are more susceptible to typhoid fever and at greater risk of typhoid ileal perforation [28, 29], which may explain the higher rate of deaths attributed to fever without any indication of organ involvement among males in Milan in 1629.

Typhoid fever is a systemic disease and consequently may also have been the cause of some of the cases of death due to fever with organ involvement reported in the MML but not included in our analysis of the epidemic. However, reports from the pre-antibiotic era suggest that gastrointestinal involvement with diarrhea occur less frequently than constipation in subjects with typhoid fever [30], and subsequent reports show that diarrhea is more common during the early phases of the disease in young children than at other ages [31]. Our study shows that deaths due to fever with gastrointestinal symptoms followed the trend of the described epidemic, but a specific analysis of such deaths did not show a significant difference in distribution by age group (perhaps because of the limited number of cases and inaccuracies in indicating the causes of death in children). Moreover, although a cough and sore throat may appear before the onset of fever in subjects with typhoid fever, pneumonia is generally infrequent and more commonly occurs in children [32]. The distribution of deaths due to fever with respiratory symptoms in our casefile is characterised by two modest peaks in the first and last months of the year, which suggests a possible association with seasonal flu and a general trend that not related to the described epidemic. Finally, typhoid fever may be associated in some cases with sparse rose spots on the trunk (the so-called typhoid roseolas) that can be subtle and short lived, so easily escaping physical examination [33].

Seventeenth century Milan was intersected by numerous canals, had a rich and easily accessible groundwater table, and its drinking water came from thousands of private household wells that were often dangerously close to cesspits containing the excreta of its inhabitants [34]. The chronicles report that the summer of 1629 was very hot and without rain for three months [2, pages 20–23], a factor that could lead to water scarcity and force people to drink contaminated water.

Poor sanitation and water supplies are still the main causes of enteric fever throughout the world and accounted for more than 14 million cases in 2017 [25]. In addition to typhoid fever,

other orofecal transmissible bacteria (non-typhoidal *Salmonella* spp., *Shigella* spp., *Escherichia coli*) and viruses (enterovirus, norovirus, adenovirus) may cause enteric fevers in hot climates. However, the infections caused by these agents are prevalently characterised by gastrointestinal symptoms (vomiting and diarrhea) and with a case fatality rate generally lower than that of typhoid fever. Furthermore, although the symptoms may be severe in infants, the elderly, and debilitated individuals, the clinical characteristics of these infections are quite different from those of the 1629 fever epidemic.

In the pre-antibiotic era, the lethality of typhoid fever was more than 10% [30], and mortality rates of as high as 30–50% are still reported in countries in which severe typhoid fever is endemic, such as Papua New Guinea and Indonesia [35]. It should also be noted that health and social welfare conditions were dramatically compromised in Milan in 1629, as is borne out by the 118 residents recorded as dying of starvation.

This study has some limitations in addition to those due to the inevitable difficulties involved when trying to attribute a cause to an epidemic solely on the basis of written documents and without any investigation of human remains. The first is the lack of information concerning the deaths occurring in the Lazaretto and other hospitals. Secondly, it is impossible to assess the fever-related deaths occurring during childhood because of the widespread use of the convenient diagnosis of *epilepsia* in 340 cases among those aged less than eight years. The physicians of the time used the term *epilepsia* to refer to infantile weakness not necessarily accompanied by febrile convulsions [36]: it was frequently used in this sense in the MML because it was a generic diagnosis that was sufficient to rule out the suspicion of plague. Thirdly, the cause of death could not always be deduced because the available information was insufficient. Finally, the lack of information concerning all the deaths that occurred in the city and the precise number of inhabitants at the time of the epidemic prevents any estimate of mortality in general and by specific cause. However, given that the available estimates indicate a total population of about 130,000 inhabitants [37] and a typhoid fever lethality rate of about 10% [30], it can be supposed that no less than 30,000 people (more or less a quarter of the population) were affected by the disease, enough to justify the alarm raised.

In conclusion, a serious epidemic probably due to enteric fever (possibly typhoid fever) occurred in Milan during the summer of 1629, and was favoured by the poor economic and nutritional conditions of a significant proportion of the population. It is possible that the concomitant existence of famine, war, and the arrival of many impoverished peasants from the surrounding countryside caused an extensive epidemic, but this was not unusual and had been largely ignored in previous years. The subsequent arrival of the Great Plague of 1630, which is estimated to have caused tens of thousands of deaths, almost completely erased all memory of the *morbino* of 1629.

## Supporting information

**S1 File. Data set constructed to perform the epidemiological investigation.**
(XLSX)

**S1 Fig. Monthly distribution of deaths due to the different types of fever.**
(JPG)

## Author Contributions

**Conceptualization:** Massimo Galli, Spinello Antinori, Folco Vaglienti.

**Data curation:** Massimo Galli, Letizia Oreni, Anna Lisa Ridolfo, Angelo Formenti.

**Formal analysis:** Letizia Oreni, Ester Luconi, Patrizia Boracchi, Elia Biganzoli.

**Investigation:** Massimo Galli, Letizia Oreni, Anna Lisa Ridolfo.

**Methodology:** Massimo Galli, Letizia Oreni, Elia Biganzoli.

**Supervision:** Folco Vaglienti.

**Validation:** Massimo Galli, Letizia Oreni, Elia Biganzoli.

**Visualization:** Letizia Oreni, Anna Lisa Ridolfo.

**Writing – original draft:** Massimo Galli, Anna Lisa Ridolfo.

**Writing – review & editing:** Massimo Galli, Anna Lisa Ridolfo, Spinello Antinori.

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
