## [Decision Letter · Decision Letter 0]

25 Jan 2023

PONE-D-22-33061

MILAN’S FORGOTTEN EPIDEMIC OF SUMMER 1629, A FEW MONTHS BEFORE THE LAST GREAT PLAGUE: AN INVESTIGATION INTO THE POSSIBLE CAUSE.

PLOS ONE

Dear Dr. Galli,

Thank you for submitting your manuscript to PLOS ONE. After careful consideration, we feel that it has merit but does not fully meet PLOS ONE’s publication criteria as it currently stands. Therefore, we invite you to submit a revised version of the manuscript that addresses the points raised during the review process.

Thank you very much for submitting your manuscript "MILAN’S FORGOTTEN EPIDEMIC OF SUMMER 1629, A FEW MONTHS BEFORE THE LAST GREAT PLAGUE: AN INVESTIGATION INTO THE POSSIBLE CAUSE" for consideration at PLOS Neglected Tropical Diseases. As with all papers reviewed by the journal, your manuscript was reviewed by members of the editorial board and by several independent reviewers. In light of the reviews (below this email), we would like to invite the resubmission of a significantly-revised version that takes into account the reviewers' comments.

We cannot make any decision about publication until we have seen the revised manuscript and your response to the reviewers' comments. Your revised manuscript is also likely to be sent to reviewers for further evaluation.

We look forward to receiving your revised manuscript.

Kind regards,

Andréa Sobral

Academic Editor

PLOS ONE

Journal Requirements:

4. Please amend the manuscript submission data (via Edit Submission) to include author Folco Vaglienti.

Reviewers' comments:

Reviewer's Responses to Questions

**Comments to the Author**

1. Is the manuscript technically sound, and do the data support the conclusions?

Reviewer #1: Yes

Reviewer #2: Partly

Reviewer #3: Partly

2. Has the statistical analysis been performed appropriately and rigorously? 

Reviewer #1: Yes

Reviewer #2: Yes

Reviewer #3: Yes

3. Have the authors made all data underlying the findings in their manuscript fully available?

Reviewer #1: Yes

Reviewer #2: Yes

Reviewer #3: Yes

4. Is the manuscript presented in an intelligible fashion and written in standard English?

Reviewer #1: Yes

Reviewer #2: Yes

Reviewer #3: Yes

5. Review Comments to the Author

Reviewer #1: Interesting article, within the scope of the journal and with valuable information about the past so that we can add information and better anticipate future epidemics, especially in the currect days where large epidemics are the reality more than the exception.

The authors used information collected at the time of the occurrence of the possible epidemic and, therefore, there is inherent difficulty in analyzing the data. There is great precariousness in the existing data in the analyzed record, justifiable by the historical component of the collection. Overall, the authors did a good job discussing the limitations of the study.

As a contribution to the discussion, perhaps the information presented could be evaluated both retrospectively for the years 1601-1628 and prospectively for the years of the "plague" period. The resulting comparisons may reinforce the idea that it was another disease and not the beginning of the plague epidemic. For example, how was the distribution of death certificates in the three periods between cerusici and physici collegiati? This seems to be the biggest source of bias with misclassification, largely due to procedures at the time that cannot be corrected. Perhaps the comparison between the three periods could show with greater certainty that it was another disease, although the symptoms described make it difficult to say which disease it would have been.

Some minor details: the access link to figure 1 is wrong; the information in figure 4 (line 119) about the sexes is not in the figure.

Reviewer #2: It was a great pleasure to review the article by Galli and collaborators, both because of the historical aspect of the paper and the interest still present in the authors of learning from the past.

Realizing we still have many things to discover, looking behind is essential to build a solid future. It was exciting to learn about ancient strategies to report or register disease cases, to look at the literature available at the beginning of the 17th century and the effort to name and classify medical findings.

However, I have some points to raise. First, I would like to check the number of fever cases between males and females presented in the abstract. Authors say, "These deaths involved 1,627 males and 1,334 females" - the numbers, in reality, refer to the number of deaths explicitly related to people with fever without any described organ involvement and not to the total of deaths.

Looking at the data collected, it's enjoyable to observe the different terms used by attending professionals according to their educational level (cerusici and phisici collegiate) and to conclude that we can still find similar inconsistencies if we look deeply into present databases. In this sense, in 1629, the lack of information under notification was understandable. What is not acceptable is that the same, for different reasons, still happens today.

Also critical, regarding the article discussion, is the need for a more profound thought in other than Typhoid fever (TF) as the most probable mortality cause. Even though the living conditions, poverty, and unsafety of food and drinking water leads us to the high probability of TF and historical evidence, it's important to look to secondary data provided by the reviewed documents: firstly, only 141 people were registered as having fever associated with gastrointestinal symptoms, only 12 presented skin rash, and 2964 have any described organ involvement. GI symptoms and skin rash are not rare in TF, and their presence should have increased the fever associated with gastrointestinal symptoms and fever with rash. Respiratory symptoms are also frequent, mainly cough, and only 144 deaths were attributed to fever with respiratory symptoms.

In the absence of more detailed disease information other than the distinct names used to register, extensively discussed by authors, it would be essential to go through infectious diseases that would lead us to a differential diagnosis in present similar cases, illnesses linked to compromised health and social welfare conditions like viral infections other than smallpox, that predominantly occur as summer epidemics.

Reviewer #3: The paper addresses a poorly studied epidemic that occurred in the seventeenth century in Milan. The paper presents unpublished data and interesting interpretations about this epidemic. The study focuses on clinical aspects, even considering the scarcity of diagnostic resources and the inadequate classification of symptoms. Besides, it brings socio-demographic elements that allow an approximation of the magnitude and dynamics of the transmission of the 'fever' that has caused thousands of deaths. On the first aspect, I recommend the review of the article by a clinical doctor, with experience in medieval medicine and the clinical paradigms of the time, such as Galenus and others.

About the second aspect, demographic and socio-epidemiological, I believe that the article can be improved by comparing estimates of the number of deaths. According to the authors "the number of deaths estimated by the chroniclers of the time is much larger than that recorded in the MML:

- 8,750 deaths according to Tadino report,

- “seven thousand souls have died”, according to the Tuscany resident,

- 1,880 (5,993 - 4,113) deaths estimated by excess mortality,

- 3,363 deaths “attributed to, or associated with fever”, according to the MML registers.

The first question is whether some deaths may have been omitted from the MML's archives, for example, residents outside the Milan´s walls, jews in ghettos, migrant populations, besides people in hospitals and lazarettos.

The second issue concerns the comparison between previous epidemics by means of metrics, some of which are already available in the article, such as median age of deaths, sex ratio, days from symptoms offset to death, and seasonality. Would these metrics be different for the black plague epidemic? Other outbreaks of fecal-oral transmission diseases? See, for example: Morabia (2009) doi:10.1017/S0950268809990136 and Bakach et al. (2015) https://doi.org/10.1093/trstmh/trv075.

In this sense, a more accurate description of climatic, political, and demographic events, briefly mentioned by the authors, that took place in 1629 to 1630 could bring a major contribution to the text, using as information sources some chronicles and reports of the time.

These suggestions do not require new calculations, but rather short supplementary texts, with reference to some historical documents and published review articles.

Other minor changes:

In the sentence ‘The number of deaths recorded in 1629 was 45.7% higher than the 4,113 annual average number in the period 1601-1628 and, in comparison with this, the excess deaths occurring in 1629 started in May and peaked in July (Fig. 1)”, 'highlight which values, in which months may be significantly different than expected, based on previous years.

The sentence “As shown in Figure 2, there were no significant differences in the distribution of deaths by gender or age group” does not match the graph. Please correct. The graph shows that “there were no significant differences in the gender distribution of deaths among age groups”.

Improve the legend of Figures 2 and 4 to explain the difference between them. What is the meaning of the numbers next to the horizontal bars? The tallest bar on the right (males) has a value of 1, and is much larger than the next bar, with a value of 6.

6. PLOS authors have the option to publish the peer review history of their article (what does this mean?). If published, this will include your full peer review and any attached files.

Reviewer #1: No

Reviewer #2: No

Reviewer #3: No

---

## [Author Response · Author response to Decision Letter 0]

24 Mar 2023

Reviewer #1. 

We are very grateful to the reviewer for his/her time and constructive comments/suggestions. 

Our replies are as follows: 

Reviewer #1

Comment 1: As a contribution to the discussion, perhaps the information presented could be evaluated both retrospectively for the years 1601-1628 and prospectively for the years of the "plague" period. The resulting comparisons may reinforce the idea that it was another disease and not the beginning of the plague epidemic. For example, how was the distribution of death certificates in the three periods between cerusici and physici collegiati? This seems to be the biggest source of bias with misclassification, largely due to procedures at the time that cannot be corrected. Perhaps the comparison between the three periods could show with greater certainty that it was another disease, although the symptoms described make it difficult to say which disease it would have been.

Reply: It has taken over a year for our team to enter, analyse and interpret the data relating only to the deaths recorded in 1629. The registers are written in Latin in different forms of handwriting and contain numerous conventional abbreviations that need to be interpreted by expert palaeographers. We are currently working on the registers of 1600-1630, but we estimate that it will take us about six months of work to complete each single year. We have therefore used the aggregated data of the registers as a reference for the first decades of the 17th century (i.e the total number of deaths occurring each year), which were published by Giuseppe Ferrario in 1840 (Fig 1). In the only available study of the registers of this period (1606 and 1607), a single non-graduate surgeon (chirurgo di sanità) called Agostino Scarparri (who was no longer active in 1629) drew up 77% of the 6869 death certificates, and the rest were drawn up by 34 different doctors, of whom 26 were defined as 'collegiate' [Dante E. Zanetti, La morte a Milano nei secoli XVI-XVIII, appunti per una ricerca. Rivista Storica Italiana, Napoli, Fascicolo I, Anno LXXXVIII, page 818]. Numerous other people became certifiers during the explosive phase of the plague epidemic in 1630, including barbers and parish elders. 

We have added a paragraph to the discussion (page 12, lines 202-218) that describes everything that reinforces our hypothesis that the 1629 epidemic was another disease and not the beginning of the plague epidemic. 

Comment 2. Some minor details: the access link to figure 1 is wrong; the information in figure 4 (line 119) about the sexes is not in the figure.

Reply: We have corrected the description of the data regarding the distribution of fever without any organ involvement by sex and age (page. 9, lines 145-146). 

 

Reviewer #2

We are very grateful to the reviewer for his/her constructive comments and suggestions. 

Our replies to comments are as follows:

Comment 1. First, I would like to check the number of fever cases between males and females presented in the abstract. Authors say, "These deaths involved 1,627 males and 1,334 females" - the numbers, in reality, refer to the number of deaths explicitly related to people with fever without any described organ involvement and not to the total of deaths.

Reply: The data concerning the deaths of males and females are now more clearly described in the abstract (lines 15-17).

Comment 2. Also critical, regarding the article discussion, is the need for a more profound thought in other than Typhoid fever (TF) as the most probable mortality cause. Even though the living conditions, poverty, and unsafety of food and drinking water leads us to the high probability of TF and historical evidence, it's important to look to secondary data provided by the reviewed documents: firstly, only 141 people were registered as having fever associated with gastrointestinal symptoms, only 12 presented skin rash, and 2964 have any described organ involvement. GI symptoms and skin rash are not rare in TF, and their presence should have increased the fever associated with gastrointestinal symptoms and fever with rash. Respiratory symptoms are also frequent, mainly cough, and only 144 deaths were attributed to fever with respiratory symptoms.

Reply: As suggested by the reviewer, we have added a paragraph concerning typhoid fever to the discussion (page 13, lines 246-261)

Comment 3. In the absence of more detailed disease information other than the distinct names used to register, extensively discussed by authors, it would be essential to go through infectious diseases that would lead us to a differential diagnosis in present similar cases, illnesses linked to compromised health and social welfare conditions like viral infections other than smallpox, that predominantly occur as summer epidemics.

Reply: The discussion now includes a paragraph concerning other pathogens that should be considered in the differential diagnosis of epidemic fevers associated with compromised health and social welfare conditions, especially during the summer season (page 14, lines 268-275).

Reviewer #3

We are grateful to the reviewer for his/her constructive comments and suggestions. 

Our replies are as follows:

Comment 1. On the first aspect, I recommend the review of the article by a clinical doctor, with experience in medieval medicine and the clinical paradigms of the time, such as Galenus and others.

Reply: The manuscript has been fully reviewed by an expert in the history of epidemics and the evolution of medical culture over the centuries.

Comment 2. About the second aspect, demographic and socio epidemiological, I believe that the article can be improved by comparing estimates of the number of deaths. According to the authors "the number of deaths estimated by the chroniclers of the time is much larger than that recorded in the MML:

- 8,750 deaths according to Tadino report,

- “seven thousand souls have died”, according to the Tuscany resident,

- 1,880 (5,993 - 4,113) deaths estimated by excess mortality,

- 3,363 deaths “attributed to, or associated with fever”, according to the MML registers.

The first question is whether some deaths may have been omitted from the MML's archives, for example, residents outside the Milan´s walls, jews in ghettos, migrant populations, besides people in hospitals and lazarettos.

Reply: a) The Methods section now more clearly explains that, by order of the health authority, registration in the MML was extended to all deaths occurring in the city except for those occurring in hospitals, convents, and the Lazzaretto, and that it was mandatory to obtain the burial permit [Vaglienti, ref. 12 ] (page 4, lines 56-58). It is therefore highly unlikely that any death was omitted. Even the deaths of the most illustrious people were recorded: for example, Alfonso Visconti, the Vicario di Provvisione (a sort of city mayor), who died in 1629, and Cardinal Archbishop Federigo Borromeo, whose death was recorded in 1631. The deaths of foreigner people were also recorded in the MML (page 5, lines 82-83 and page 10, lines 160-164). 

b) It is possible that the difference between the number of deaths recorded in the MM and those reported by chronicles or contemporary witnesses is due to the deaths occurring in hospitals (particularly the Lazzaretto) about which no documentation is available. However, it is more likely that the number of deaths reported in the ancient chronicles was inflated because of the emotional reactions of the writers and/or as a means of sounding the alarm. These aspects are discussed on page 10, lines 154- 159.

Comment 3. The second issue concerns the comparison between previous epidemics by means of metrics, some of which are already available in the article, such as median age of deaths, sex ratio, days from symptoms offset to death, and seasonality. Would these metrics be different for the black plague epidemic? Other outbreaks of fecal-oral transmission diseases? See, for example: Morabia (2009) doi:10.1017/S0950268809990136 and Bakach et al. (2015) https://doi.org/10.1093/trstmh/trv075.

Reply: a) No metrics concerning plague epidemics in Milan are available other than seasonality: plague outbreaks almost always peaked in the summer. Milan was spared from the black plague of 1347-48. As far as we know from the available documentation, when compared to the 11 major bubonic plague epidemics occurring in Milan between 1361 and 1630, the morbino of 1629 was certainly less devastating and probably less lethal. The case fatality rate of the bubonic plague in the pre antibiotic era ranged between 40 to 60% and was 42,8% during the bubonic plague outbreak in Glasgow in 1900 [Dean KR, Krauer F, Schmid BV. 2019 Epidemiology of a bubonic plague outbreak in Glasgow, Scotland in 1900. R. Soc. open sci. 6: 181695]. 

b) It should also be emphasised that, throughout the first thirty years of the seventeenth century, the highest number of registered deaths was recorded in the summers (Ferrario, ref. 14), thus suggesting the recurrence of outbreaks of enteric fevers in the hottest part of the year, the permanence of which may have been due to silent carriers in the population. Unfortunately, there are no data concerning these outbreaks (other than their seasonality) that would allow us to make a comparison with the metrics of the 1629 epidemic. We have discussed this on page 13, lines 232-236.

Comment 4. A more accurate description of climatic, political, and demographic events, briefly mentioned by the authors, that took place in 1629 to 1630 could bring a major contribution to the text, using as information sources some chronicles and reports of the time.

A sentence concerning the political situation in Milan and northern Italy has been added to the introduction (page 3, lines 22-29). 

Milan’s demographic data of 1629 are based on estimates (the first census was not carried out until 1861). The estimate of about 130,000 inhabitants in 1629 (page 15, line 293, Ref 37) is generally accepted although it has many limitations. 

Systematic surveys of climatic data in Milan started in1763. However, the chronicles report that the summer of 1629 was very hot and without rain for three months (Ref. 2). This information has been included in the discussion (page 14, lines 264-266).

Comment 5. In the sentence ‘The number of deaths recorded in 1629 was 45.7% higher than the 4,113 annual average number in the period 1601-1628 and, in comparison with this, the excess deaths occurring in 1629 started in May and peaked in July (Fig. 1)”, 'highlight which values, in which months may be significantly different than expected, based on previous years.

Reply: We have corrected the sentence as suggested (page 5, lines 86-89).

Comment 6. The sentence “As shown in Figure 2, there were no significant differences in the distribution of deaths by gender or age group ? does not match the graph. Please correct. The graph shows that “there were no significant differences in the gender distribution of deaths among age groups”.

Reply: We have corrected the sentence as suggested. (page 5, lines 95-96).

Comment 7. Improve the legend of Figures 2 and 4 to explain the difference between them. What is the meaning of the numbers next to the horizontal bars? The tallest bar on the right (males) has a value of 1, and is much larger than the next bar, with a value of 6.

Reply: We have improved the legend of Figure 2 by clarifying that the numbers next to the bars are the absolute number of deaths.

---

## [Editor Report · Decision Letter 1]

4 Apr 2023

MILAN’S FORGOTTEN EPIDEMIC OF SUMMER 1629, A FEW MONTHS BEFORE THE LAST GREAT PLAGUE: AN INVESTIGATION INTO THE POSSIBLE CAUSE.

PONE-D-22-33061R1

Dear Dr. Galli,

We’re pleased to inform you that your manuscript has been judged scientifically suitable for publication and will be formally accepted for publication once it meets all outstanding technical requirements.

Kind regards,

Andréa Sobral

Academic Editor

PLOS ONE

Additional Editor Comments (optional):

Thank you for your fine contribution.  On behalf of the Editors of the PLOS, we look forward to your continued contributions to the Journal.
---

## [Editor Report · Acceptance letter]

30 May 2023

PONE-D-22-33061R1 

Milan’s forgotten epidemic of summer 1629, a few months before the last great plague: an investigation into the possible cause. 

Dear Dr. Galli:

I'm pleased to inform you that your manuscript has been deemed suitable for publication in PLOS ONE. Congratulations! Your manuscript is now with our production department. 

Kind regards, 

on behalf of

Dr. Andréa Sobral 

Academic Editor

PLOS ONE